# Monocular Camera-Based Robotic Pick-and-Place in Fusion Applications



Ruochen Yin [1,2,3], Huapeng Wu [3,*], Ming Li [3], Yong Cheng [1,4], Yuntao Song [1,*] and Heikki Handroos [3]

1  Institute of Plasma Physics Chinese Academy of Sciences (ASIPP), Chinese Academy of Sciences, Hefei 230031, China
2  University of Science and Technology of China, Hefei 230026, China
3  Laboratory of Intelligent Machines, School of Energy Systems, LUT University, 53850 Lappeenranta, Finland
4  Institute of Energy, Hefei Comprehensive National Science Center, Hefei 230031, China
*  Correspondence: huapeng.wu@lut.fi (H.W.); songyt@ipp.ac.cn (Y.S.)

**Abstract:** Robotic pick-and-place represents a nascent but swiftly evolving field in automation research. Most existing research relies on three-dimensional (3D) observations obtained directly from the 3D sensor or recovered by the two-dimensional (2D) camera from multiple perspectives. In this paper, we introduce an end-to-end pick-and-place neural network that solely leverages simple yet readily accessible data, namely, monocular camera and forward kinematics, for fusion applications. Additionally, our approach relies on the deep reinforcement learning (DRL) algorithm to facilitate robots in comprehending and completing tasks. The entire process is data-driven, devoid of any artificially designed task sessions, which imbues our approach with enhanced flexibility and versatility. The proposed method exhibits excellent performance in our experiment.

**Keywords:** robotic pick-and-place; monocular camera; neural network; reinforcement learning; fusion applications

## 1. Introduction

In addressing the energy crisis, fusion energy is deemed a dependable and secure source for the future. The majority of fusion energy experimental devices worldwide depend on plasma-facing components (PFCs) to enable the device to operate under extreme conditions. These PFCs are subjected to hyper heat, pressure, and magnetic flux directly. As such, the periodic maintenance and replacement of these components are required. Currently, these maintenance tasks primarily depend on remote handling systems (RHSs) that require high expertise and are time-consuming. Thus, intelligent technology-based automated maintenance systems are crucially required for the subsequent generation of fusion reactors.

Of all the maintenance tasks, the robotic pick-and-place issues account for a considerable proportion. Despite extensive research on robotic pick-and-place over the past decade, this problem has not been entirely resolved. More complex tasks and environments are emerging, while the performance demands for robotic pick-and-place are escalating, as exemplified by the robotic pick-and-place operation in a fusion reactor that faces challenging environments and stringent requirements.

In the realm of robotic pick-and-place, most algorithms with advanced performance rely on 3D spatial data from depth cameras or multi-view camera systems. However, inside the fusion reactor, both the PFCs and other components are made of metal with smooth surfaces. Active depth cameras, e.g., time-of-flight (TOF) cameras, utilize a technique that involves measuring the distance from the camera to the target through the computation of the time difference between when light leaves and returns. This technology works optimally on surfaces that are diffuse and possess rough, non-glossy properties. However, if the surface is excessively smooth and reflects light specularly, then light incident on the

surface at a certain angle will be deflected in another direction and will not return to the sensor. As a result, the sensor will not be able to accurately calculate the distance to the object, as the return signal cannot be received. This error is a systematic error and cannot be eliminated through statistical methods. Passive depth cameras or multi-view camera systems, e.g., stereo cameras, rely on feature point extraction methods to create correlations between images acquired by different cameras to derive depth information for each pixel. Then, as mentioned above, the metal material inside the fusion reactor is highly uniform in color and texture, making it difficult to extract and correlate feature points in the images.

As a solution, we propose an end-to-end pick-and-place network that solely relies on monocular RGB images and the 3D position data of the end-effector that are both available and reliable in the fusion reactor environment. There are quite number of different methods to achieve monocular camera-based robotic pick-and-place. The most common method for pose estimation is to use the convolutional neural network (CNN) to extract 2D keypoints from the image, and then solve the perspective-n-point (pnp) [1] problem based on some other parameters, e.g., camera internal. This type of method requires training data for different tasks and objectives and relies on manually designing during the task planning process, which lacks generalizability and feasibility in the fusion reactor environment.

DRL-based algorithms are an intelligent choice for creating end-to-end solutions. However, due to the specificity of reinforcement learning (RL), it can be dangerous and inefficient to deploy it directly in a realistic environment. Therefore, the most common approach is to train the model in a simulation environment and then verify it in the real world.

In this paper, we implement the distributed proximal policy optimization (DPPO) [2] algorithm using the Isaac Gym simulation engine [3] to generate a simulation environment with multiple parallel sub-environments. Moreover, all the sub-environments' training processes are rendered in real time to help us monitor the model's performance, and all these processes are accelerated by the GPU.

Compared to other monocular camera-based methods, our approach differs in several ways:

1. Unlike other monocular camera-based methods that can only place the target at a fixed location after random object picking, our method allows for completely random object picking and placement.
2. Our method does not rely on any a priori information or auxiliary methods, such as visual markers, to aid in target recognition.
3. Our approach does not divide the pick-and-place task into multiple phases, such as target recognition and task planning, and does not require an artificially designed task planning module. Rather, it is an end-to-end solution that is data driven.

Thus, the main innovations and contributions of this paper are as follows:

1. We propose a lightweight end-to-end method for handling the pick-and-place task, where all the sessions, from environmental perception to task understanding, are conducted by deep reinforcement learning without any artificially designed components. The proposed method's training can be executed on a consumer-level GPU.
2. The proposed method solely relies on RGB data from an eye-in-hand camera and the end-effector's center position, both of which are reliable and available for fusion applications.
3. We developed multiple sub-environments in the simulation environment for parallel training and rendered each sub-environment's current state in real time to facilitate adjusting hyper-parameters and accelerating the training speed.

## 2. Related Work

Robotic pick-and-place is a subfield of robotic manipulation research, which requires robots to not only recognize basic scenes but also have a high-level understanding of tasks. In recent years, with the rapid development of artificial intelligence, significant progress has been made in robotic manipulation research, with various technical approaches being developed. In this review, we will expand beyond robotic pick-and-place and also include

relevant research on robotic manipulation. Specifically, we will briefly review research on target recognition and task planning.

### 2.1. Deep Learning-Based Methods

With the remarkable achievements that deep learning (DL) and neural networks (NNs) have made in the field of target recognition, using NN-based methods is the most popular approach in robotic pick-and-place and manipulation research. These NN-based methods can be further subdivided depending on the input and output data.

### 2.1.1. 2D Image Input with Traditional Computer Vision Algorithms

Due to the availability of dense 3D data, depth cameras have become the first choice for most robotic manipulation research. Nevertheless, in certain exceptional scenarios, such as reflective surfaces with specular reflections and transparent objects, depth cameras fail to provide trustworthy observation data, as previously noted. Consequently, the monocular camera-based approach continues to occupy a significant position.

Directly acquiring 3D information for robotic manipulation tasks through monocular camera images is not possible. The key challenge in such studies lies in relying on 2D information for estimating three-dimensional pose. A common approach to tackle this challenge is to leverage deep neural networks (DNNs) to extract 2D keypoints from the image. Through calibration, camera intrinsics and joint configuration can be obtained, then the perspective-n-point (PnP) algorithm is used to recover 3D information. Building on this approach, Lee et al. [4] proposed a method to estimate robot pose using an eye-to-hand camera. This method computes camera extrinsics from a single image frame, thereby enabling online robot pose estimation. Byambaa et al. [5] proposed a similar approach to address the challenge of the robotic manipulation of transparent glass objects.

Horng et al. [6] employed fast region with CNN to recognize the object and subsequently estimate depth by leveraging the camera's field of view (FOV), as well as the k-nearest neighbors (kNN) and fuzzy inference system (FIS) algorithms.

Unlike all the above methods, Nguyen et al. [7] extracted deep features from the input video frames with a deep CNN and then introduced the time sequence information by adding recurrent neural network (RNN) structures into the network so that the 3D knowledge could be learned.

### 2.1.2. 3D input Data with CNN

The methods that employ 3D input data share the common approach of feeding observations from the 3D camera into a CNN, but the network outputs differ. In the method proposed by Zeng et al. [8], which is based on fully convolutional networks (FCNs), the NN outputs a pixel-wise grasping quality heat map for the input images. The key feature of this work is that the algorithm can select the appropriate end-effector for different types of identified objects and place it in the corresponding position.

Schwarz et al. [9] proposed a method that combines the outputs of an object classification NN and an object segmentation NN to manipulate objects in the cluster. While the method yields good results, modern techniques could potentially replace the two networks presented in the paper with an instance segmentation network. Zeng et al. put forward transporter networks [10] that utilize RGB-D information to facilitate diverse automated operations for robotic arms via learning from demonstration (LFD). This approach avoids any objectivity assumptions and displayed exceptional performance on several benchmarks in both simulation and real-world experiments.

Regardless of whether the methods use 2D image input or 3D input, the task planning parts rely on either artificial design or LFD, which can greatly impact the generalizability of the algorithm. However, this issue was addressed through the emergence and rapid development of reinforcement learning (RL) techniques.

*2.2. Deep Reinforcement Learning-Based Methods*

Unlike DL-based approaches, DRL-based methods do not involve the division of problems into sub-tasks, such as target identification or task planning. Instead, they offer an end-to-end solution by integrating RL algorithms with DNNs. This enables the entire algorithm to be driven by data, which in turn results in improved generalizability and robustness of the algorithm.

The crux of RL lies in enabling the agent/robot to explore the task space independently, rewarding it for different states and actions based on predefined guidelines, and updating the agent's behavioral strategy with the aim of maximizing rewards. However, this process is often fraught with potential dangers and can be time consuming. Therefore, it is common practice to construct a virtual environment and train the agent within it.

Quillen et al. [11] proposed a benchmark for comparing the performance of different policy-based and value-based RL algorithms for robotic grasping tasks. James et al. [12] introduced the attention-driven robotic manipulation (ARM) algorithm, which first trains a Q-attention network to extract the area of interest (AOI) from input RGB and point cloud data. The AOI data are then used to generate control commands by a next-best pose estimate network. Mohammed et al. [13] proposed an object pick-and-place algorithm for cluttered scenes, based on the Q-learning algorithm [14]. In their approach, RGB and 3D point cloud data are passed through a dense network (DenseNet) to generate pixel-wise Q-value predictions, and placing action is executed based on artificially designed behavioral logic. Taking a similar approach, Gualtieri et al. [15] employed RGB-D information and a priori knowledge regarding the target object category to train a robotic arm using DQN. This allowed the arm to carry out pick-and-place operations for bottles and water glasses within a realistic environment.

Gu et al. [16] proposed an asynchronous NAF method, based on the normalized advantage function (NAF) algorithm [17]. This method can collect observations, actions, and rewards for each time step on different agents and asynchronously update the parameters of the deep neural network Q-function approximator. In simulation experiments, the target position is fed as an observation into the neural network, which significantly reduces the task's difficulty but is challenging to achieve in reality. Lee et al. [18] presented an intriguing work that involves leveraging a multi-view vision system to train a robotic arm to solve the challenge of stacking objects of diverse shapes via reinforcement learning.

## 3. Proposed Method

Based on the above brief review, current pick-and-place algorithms for robotics primarily rely on either RGB image input data with traditional algorithms or a combination of 3D input data and DRL-based methods. The former requires a human-designed mission planning phase and lacks generalizability, while the latter is an end-to-end approach but has input data requirements that cannot be met in fusion applications.

Therefore, this paper proposes an end-to-end method that only requires an RGB image and the center position of the end-effector (CPE), both of which are available and reliable in a fusion application environment. The proposed method is described in detail below.

*3.1. Simulation Environment Establishment and Problem Statement*

In this work, we selected Isaac Gym as our simulation environment. Isaac Gym offers the capability to create an unlimited number of independently operating sub-environments and acquire the agent's observations, actions, and rewards from each sub-environment, storing the results in PyTorch [19] GPU tensors. The state of each agent is rendered in real time, and all physics simulation processes in Isaac Gym run on the GPU.

Within our simulation environment, one of the sub-environments is depicted in Figure 1. As the figure illustrates, each sub-environment comprises a six-degree-of-freedom (DOF) UR-10 robot end-mounted with an FE gripper, and two cubes that are randomly placed on a table. The task's objective is to pick up the white cube and place it onto the purple one. The

blue cube in the figure represents the eye-in-hand camera, which is attached to the robot's last wrist link.

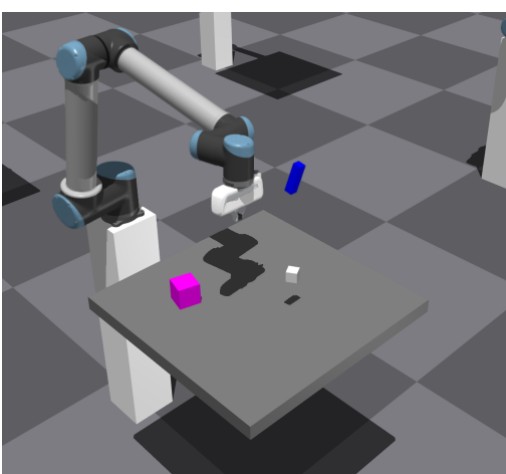

**Figure 1.** A sub-environment in our Isaac Gym simulation environment.

### 3.2. Distributed Proximal Policy Optimization Algorithms

RL is a Markov decision process (MDP), which could be represented by $(S, \mathcal{A}, \mathcal{P}, \mathcal{R}, \gamma)$. Here, $S$ represents the state space, $\mathcal{A}$ denotes the action space, $\mathcal{P}$ refers to the state transition possibility, $\mathcal{R}$ stands for the reward space, and $\gamma$ represents the discount factor. The objective of all RL algorithms is to maximize the cumulative rewards, formulated as $max \sum_{t=1}^{T} \gamma^t r_t$, where the $t$ is the time step and $r_t \in \mathcal{R}$ is the reward at step $t$.

Although the goals are the same, the accumulated rewards have different equivalent forms in different algorithms. One of the two basic forms is the state value function $V(s)$, which is defined in Formula (1):

$$
\begin{aligned}
V(s) &= E\left[\sum_{k=0}^{\infty} \gamma^t r_{t+k+1} | S_t = s\right] \\
&= E\left[r_{t+1} + \gamma r_{t+2} + \gamma^2 r_{t+3} + \dots | S_t = s\right] \\
&= E[r_{t+1} + \gamma V(S_{t+1}) | S_t = s]
\end{aligned}
\tag{1}
$$

where $S_t \in S$ is the state in time step $t$, $r \in \mathcal{R}$ is the reward, and all subscripts indicate the time step.

Another one is the state–action value function $Q(s, a)$, as shown in Formula (2):

$$
\begin{aligned}
Q_\pi(s, a) &= E_\pi\left[\sum_{k=0}^{\infty} \gamma^t r_{t+k+1} | S_t = s, A_t = a\right] \\
&= E_\pi\left[r_{t+1} + \gamma r_{t+2} + \gamma^2 r_{t+3} + \dots | S_t = s, A_t = a\right] \\
&= E_\pi[r_{t+1} + \gamma Q_\pi(S_{t+1}, A_{t+1}) | S_t = s, A_t = a]
\end{aligned}
\tag{2}
$$

In comparison to Formula (1), Formula (2) incorporates an additional variable $\pi \in \mathcal{P}$, which represents the policy. The simplest policy is to employ the greedy algorithm, which sets $\pi(s) = arg \max_a Q(s, a)$. However, to strike a balance between exploration and exploitation, a more intricate policy must be utilized. Furthermore, the following equations demonstrate the relationship between $Q(s, a)$ and $V(s)$:

$$
\begin{aligned}
V(s) &= \sum_{a \in \mathcal{A}} \pi(a|s) Q_\pi(s, a) \\
Q_\pi(s, a) &= r_s^a + \gamma \sum_{s' \in S} \mathcal{P}_{ss'}^a V(s')
\end{aligned}
\tag{3}
$$

where the $s'$ is the new state, $\pi(a|s)$ is the policy and $\mathcal{P}^a_{ss'}$ is the probability that the agent takes action $a$ in the current state $s$ and moves to the new state $s'$.

Thus, RL algorithms can be divided into value-based and policy-based groups. The value-based algorithms focus on $Q(s,a)$ and guide the agent to the final goal by updating the $Q(s,a)$ value with a fixed policy, such as the previously mentioned greedy strategies. The policy-based algorithms modify the actions' probability, e.g., $\mathcal{P}^a_{ss'}$ in Formula (3) by parameterizing and updating the policy function $\pi(a|s)$ to ensure that the agent obtains the maximum accumulated rewards.

The two aforementioned methodologies possess their respective merits and demerits. Generally speaking, value-based algorithms tend to converge at a slower rate, but they have the ability to reach the global optimal solution. Additionally, they tend to exhibit subpar performance when dealing with continuous space problems. Conversely, policy-based algorithms tend to converge at a faster rate; however, during later stages, these algorithms tend to oscillate around the optimal value function with minimal fluctuations and are unable to attain the optimal value result.

Therefore, an important DRL algorithm called advantage actor–critic (A2C) [20] which depends on the actor–critic [21] is presented. A2C combines the value function and policy together, the actor decides which action to take, and the critic tells the actor how good its action was and how it should adjust. The brief process is shown in Algorithm 1.

---

**Algorithm 1** A2C.

1: Take action $a \sim \pi_\theta(a|s)$, get $\{s, a, r, s'\}$
2: Update $\hat{v}^\pi_\phi(s)$ using target $r + \gamma\hat{v}^\pi(s')$ {The function approximation idea is used here, $\hat{v}^\pi_\phi(s)$ is the approximation of $\hat{v}^\pi(s)$}
3: Evaluate $\hat{A}^\pi(s_i, a_i) = r(s_i, a_i) + \gamma\hat{v}^\pi_\phi(s') - \hat{v}^\pi_\phi(s)$
4: $\nabla J(\theta) = \nabla_\theta \log \pi_\theta(a_i|s_i)\hat{A}^\pi(s_i, a_i)$
5: $\theta \leftarrow \theta + a\nabla_\theta J(\theta)$

---

The A2C algorithm excels at managing continuous control problems. However, it inherits the drawback of policy-based algorithms, which makes it challenging to converge to the global optimal solution. The primary reason for this challenge is the difficulty in determining the appropriate step size/learning rate during the policy gradient. To address this issue, the proximal policy optimization (PPO) [22] algorithm is introduced based on the idea of trust region policy optimization (TRPO). The objective of TRPO is illustrated below:

$$Maximize_\theta E_s \left[ \frac{\pi_\theta(a|s)}{\pi_{\theta old}(a|s)} \hat{A} \right] \tag{4}$$

To limit the difference between the new policy and the old one, TRPO introduces the Kullback–Leibler (KL) divergence constraint:

$$E_t[KL(\pi_{\theta old}(\cdot|s), \pi_\theta(\cdot|s))] \le \delta \tag{5}$$

where $\delta$ is the size of the constraint region, then the final target function of the actor turns into

$$Maximize_\theta E_s \left[ \frac{\pi_\theta(a|s)}{\pi_{\theta old}(a|s)} \hat{A} - \beta KL[\pi_{\theta old}(\cdot|s), \pi_\theta(\cdot|s)] \right] \tag{6}$$

where the $\beta$ is the fixed penalty coefficient. However, research revealed that the KL penalty update performs less effectively in reality compared to the clipping option, as illustrated below:

$$L^{CLIP}(\theta) = E_t \left[ \min(r_t(\theta)\hat{A}, clip(r_t(\theta), 1 - \epsilon, 1 + \epsilon)\hat{A}) \right] \tag{7}$$

To summarize, the PPO algorithm can be presented as Algorithm 2. On the other hand, the distributed proximal policy optimization (DPPO) algorithms employ multiple threads based on PPO to gather state, action, reward, and other data from numerous agents trained in a distributed manner. These data are then utilized to update parameters synchronously. While the theoretical model is consistent with PPO, the challenge lies primarily in the engineering implementation.

---

**Algorithm 2** Proximal policy optimization (PPO).

---

1: Run policy $\pi_\theta$ for $T$ timesteps, collecting $\{s_t, a_t, r_t\}$
2: Estimate advantages $\hat{A}_t = \sum_{t'>t} \gamma^{t'-t} r_{t'} - V_\phi(s_t)$
3: Update the policy by PPO-Clip objective:
  $\theta_{k+1} = \arg\max_\theta \sum_{t=0}^{T} min(R_t(\theta)\hat{A}_t, \, clip(R_t(\theta), 1-\epsilon, 1+\epsilon)\hat{A}_t)$, where $R_t(\theta) = \frac{\pi_\theta(a_t|s_t)}{\pi_{old}(a_t|s_t)}$
4: Updating the value function:
  $\phi_{k+1} = \arg\min \sum_{t=0}^{T} (\sum_{t'=t}^{T} \gamma^{t'-t} r_{t'} - V_\phi(s_t))^2$

---

*3.3. Network Architecture*

In our work, we combine the RGB image with the 3D position of the CPE; thus, the network should have two input branches. The architecture of the network is as shown in Figure 2.

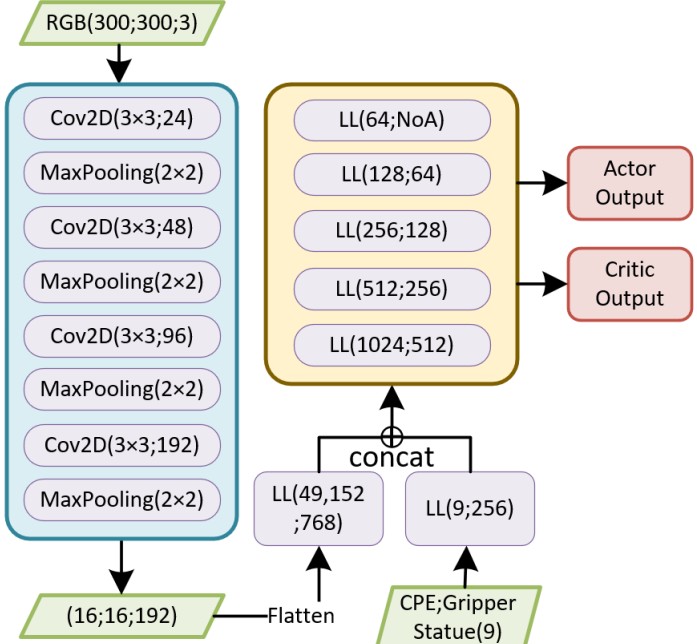

**Figure 2.** The architecture of the network. In the figure, *FC* indicates fully connected layer, and *NoA* indicates number of actions.

It is noteworthy that the network exclusively features one output. The rationale behind displaying actor output and critic output in the figure pertains to the existence of an actor network and a critic network, respectively. Both networks possess an identical structure but differ in terms of their parameters. Critic output is representative of the value, while actor output pertains to the policy, specifically the KL divergence of the policy. With the exception of the last layer, all fully connected layers in the network leverage the Elu activation function [23]. Furthermore, the Adam optimizer [24] was chosen for this network.

### 3.4. Reward Setting

In a RL method, the reward need to be carefully designed. An appropriate reward design can keep the agent's 'curiosity' and guide it to the final goal. In our work, we design a variety of rewards according to the task process, as shown in Algorithm 3.

---

**Algorithm 3** Reward setting.

1: Getting points $P_a, P_b, \ldots, P_e$ from Figure 3, $P_k(x_k, y_k, z_k)$ is the 3D coordinates of point $k \in \{a, b, c, d, e\}$, $D_{ab} = \sqrt{(x_a - x_b)^2 + (y_a - y_b)^2 + (z_a - z_b)^2}$ , $X_{ab} = x_a - x_b$, and so on for $Y_{ab}$ and $Z_{ab}$, $tanh(x) = (e^x - e^{-x})/(e^x + e^{-x})$, $S_a$ and $S_b$ are the side length of white cube and purple cube.

2: At the grasping stage:
    $r_{dis} = 1 - tanh(\alpha(D_{ac} + D_{ad} + 3D_{ae}))$, $\alpha$ is the weight.

3: $lifted = clip(((Z_a - Z_t - \frac{S_a}{2})/S_b), 0, 1)$, where $Z_t$ is the height of the table, $r_{lif} = lifted$.

4: $Ali_{ab} = \sqrt{(x_a - x_b)^2 + (y_a - y_b)^2 + (z_a - z_b + \frac{S_a + S_b}{2})^2}$

5: **if** $lifted > 0.5$ **then**

6:     $r_{ali} = 1 - tanh(\gamma Ali_{ab})$, $\gamma$ is the weight;

7: **else**

8:     $r_{ali} = 0$

9: **end if**

10: $r_{dis} = max(r_{dis}, r_{ali})$

11: **if** $\sqrt{X_{ab}^2 + Y_{ab}^2} < 0.02$ and $|z_a - S_b - S_a/2| < 0.01$ **then**

12:     $r_{sta} = clip(D_{ae}, 0, 1)$

13: **else**

14:     $r_{sta} = 0$

15: **end if**

16: $r_{dis} = max(r_{dis}, r_{sta})$

17: The final reward $R = w_1 r_{dis} + w_2 r_{lif} + w_3 r_{ali} + w_4 r_{sta}$, $w_1, \ldots, w_4$ are the weights.

---

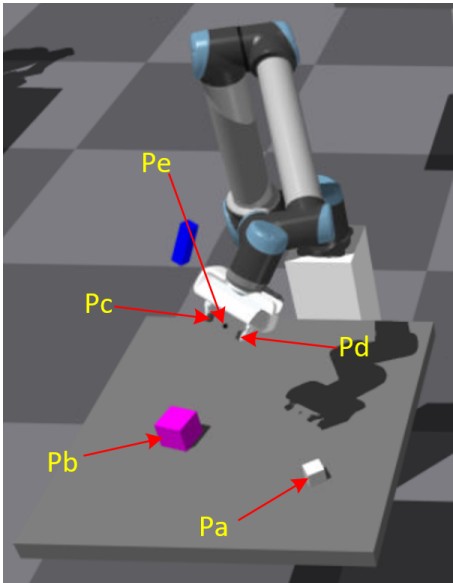

**Figure 3.** Some important positions in our task, in this picture, $P_a$ and $P_b$ are the center points of the two cubes, $P_c$ and $P_d$ are the tips of two fingers and $P_e$ is the CPE.

## 4. Experiment

We carried out our experiment on a computer with a Nvidia RTX 2080Ti GPU (12 Gigabyte Memory), Intel i7-10700k CPU, and 64GB of RAM. We created 30 sub-

environments in the simulation, as shown in Figure 4. Our code is available at here (https://github.com/rolandying/Monocular-based-pick-and-place, accessed on 29 March 2023). The implementation of some algorithms depends on the RL_games [25] package.

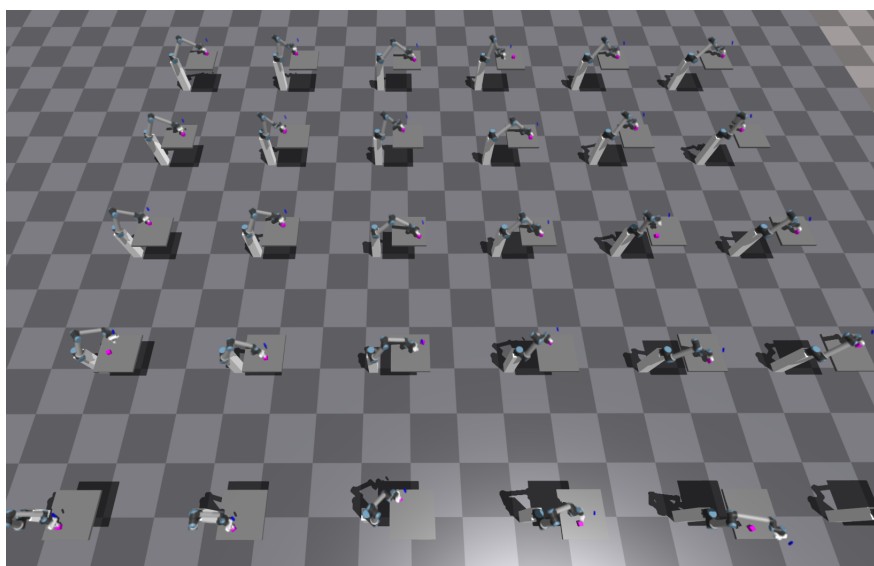

**Figure 4.** The simulation environment, the number of the sub-environment is depend on the size of GPU Memory.

Figure 5 illustrates the reward values during the training process. Following an initial bottleneck phase, the reward value experiences a remarkable surge at approximately 4000 epochs and continues to increase thereafter. This corresponds to the experimental findings that the robot is able to proficiently grasp a target placed randomly on the table following the first 4000 epochs of training. Subsequently, the alignment and stacking processes commence, which take longer, primarily because an eye-in-hand camera is utilized. When the robot arm grasps the white cube, the camera's limited field of view prevents direct observation of the purple cube. Consequently, the robot must rely on historical observation data to ascertain the subsequent destination, resulting in a lengthier learning process.

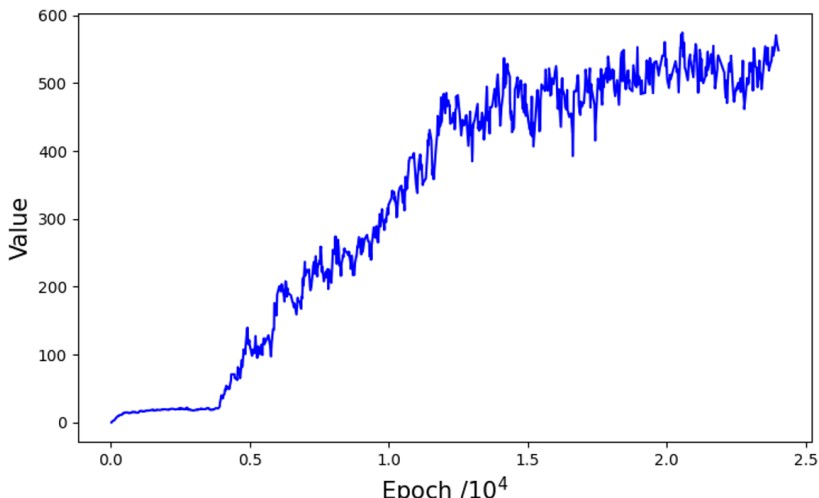

**Figure 5.** The reward during the training.

The losses incurred during training are depicted in Figure 6. Analysis of the figure indicates that the value of actor losses (AL) consistently oscillates within a confined range,

signifying a stable policy change ratio. This underscores the efficacy of the clipping operation. The value of critic losses (CL) exhibits a continuous decline and ultimately stabilizes, except for a single abrupt change in the middle. At roughly 4 million steps, both AL and CL values experience a peak. This aligns precisely with the moment in Figure 5 when the robot makes a significant breakthrough and adeptly grasps the target around the 4000th epoch. At this point in time, both the value and policy undergo a rapid transformation.

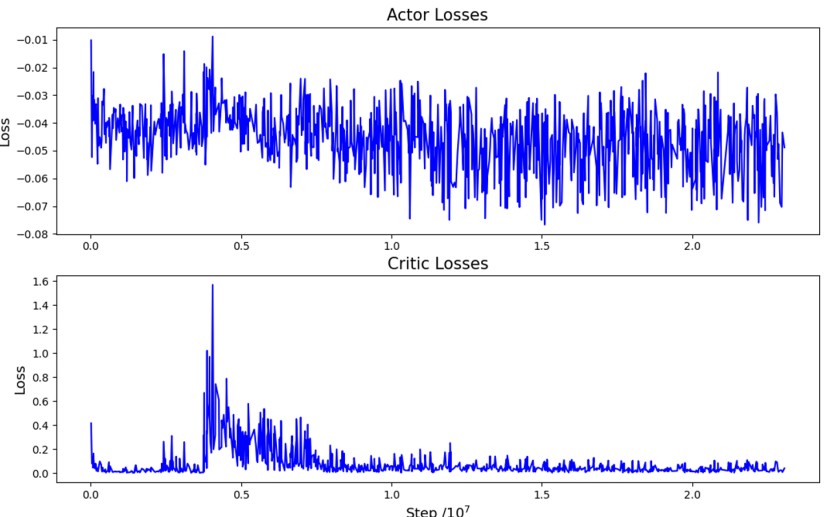

**Figure 6.** Actor losses and critic losses during the training.

We conducted a comparative analysis between our monocular method and the multi-layer perception machine (MLP) based method. In the latter, the 3D coordinates of the centers of the two cubes are employed as input data, rendering it immensely advantageous, albeit impractical to implement in reality. The outcomes are documented in Table 1. Our method demonstrated that the agent encountered a total of three failures during the testing phase, with one of them occurring during the grasping phase and the remaining during the stacking phase. The failure during the grasping phase can be attributed to the cubes being too close to each other, thereby rendering it arduous for the robot to identify an optimal grasping pose. On the other hand, during the stacking phase, the agent primarily failed due to the target cube rolling off the destination cube, owing to poor posing when the gripper was released. For the MLP method, the agent failed once in the alignment phase and once in the stacking phase. The reason for the failure during the stacking phase remained consistent with our method, whereas the failure during the alignment phase was due to the agent unintentionally pushing the purple cube off the table. The comparison indicates that the performance of our method is marginally inferior to the MLP-based method. However, it is still remarkable, particularly considering that our method relies solely on a monocular camera. Owing to the strong advantage of the MLP method in target localization, the number of steps required is significantly lower than that of our method and the reward is slightly higher than that of our method. The entire training process took roughly 70 h.

**Table 1.** Comparison between our method and the MLP-based method.

|  | Success No./Experiment No. | | | Steps | Reward |
|---|---|---|---|---|---|
|  | Grasp | Lift & Align | Stack | NA | NA |
| Ours | 149/150 | 149/150 | 147/150 | 125.7 | 541.8 |
| MLP | 150/150 | 149/150 | 148/150 | 51.3 | 577.5 |

## 5. Conclusions

During this study, we presented a novel approach for solving the robotic pick-and-place task based on deep reinforcement learning (DRL). Our approach relies solely on a monocular camera and CPE, and it optimizes the time difference (TD) using the DRL algorithm to recover 3D information, allowing our method to meet the stringent requirements of fusion applications. The proposed method achieved outstanding performance in a simulation environment. Although we did not validate our method in a real-world scenario, we believe that the accessibility of the input data and the efficient distributed training of multiple sub-environments will facilitate its transfer to a realistic environment. Therefore, migrating the algorithm to a realistic environment and verifying its effectiveness is our next step. The key contribution of this study is showcasing the potential of using DRL algorithms with monocular cameras to recover 3D spatial information and handle related tasks with outstanding performance.

**Author Contributions:** Conceptualization, R.Y. and H.W.; Methodology, R.Y. and H.W.; Software, R.Y. and M.L.; Validation, R.Y. and H.W.; Formal analysis, R.Y.; Investigation, R.Y. and H.W.; Resources, R.Y. and M.L.; Data curation, R.Y. and Y.C.; Writing—original draft, R.Y.; Writing—review & editing, H.W. and Y.S.; Supervision, H.W., Y.C. and H.H.; Project administration, H.W.; Funding acquisition, H.W. All authors have read and agreed to the published version of the manuscript.

**Funding:** This work was carried out within the framework of the EUROfusion Consortium, funded by the European Union via the Euratom Research and Training Program (Grant Agreement No. 101052200—EUROfusion). Views and opinions expressed are, however, those of the author(s) only and do not necessarily reflect those of the European Union or the European Commission. Neither the European Union nor the European Commission can be held responsible for them. This work is also supported by the Comprehensive Research Facility for Fusion Technology Program of China Under Contract No. 2018-000052-73-01-001228 and the University Synergy Innovation Program of Anhui Province with Grant No. GXXT-2020-010.

**Institutional Review Board Statement:** Not applicable.

**Informed Consent Statement:** Not applicable.

**Data Availability Statement:** Not applicable.

**Conflicts of Interest:** The authors declare no conflict of interest.

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
