# Peer review of "Monocular Camera-Based Robotic Pick-and-Place in Fusion Applications"

_applsci, doi:10.3390/app13074487_

Round 1
Reviewer 1 Report
In the paper, you wrote that the results are only from the simulator, which is fine, but greater significance would be achieved if you tested the algorithm on at least one manipulator. I suggest this refinement.
It is not clear from the description, whether other shapes besides the cubes used here are automatically recognized, or whether they need retraining.
Author Response
The response is in the file.

Reviewer 2 Report
The authors introduce an end-to-end pick-and-place neural network that solely leverages simple yet readily accessible data, namely monocular camera and forward kinematics, for fusion applications. The idea of the paper is interesting and the paper is well writen, with some minor errors found and listed in sequence. The authors must pay careful attention to some points in order to improve paper quality and also readers's understanding of the proposed work.
The authors must improve the amount of details in the paper so that it can increase its reproducibility factor. In its current version, it would be pretty hard for a reader to reproduce the work done due to valuable missing information. A good thing to do that helps reproducibility is to make available important files in a public repository. For instance, training and evaluation scripts as long with scene files and others. What was the robotic arm model used in the simulations?
Some important references are missing in the paper:
- Transporter Networks: Rearranging the Visual World for Robotic Manipulation
- Pick and Place Without Geometric Object Models
- Beyond Pick-and-Place: Tackling Robotic Stacking of Diverse Shapes
How is the information from the sub-environments synchronized so that a virtual robot contributes to the learning of another one? This is not clear in the text.
The work proposed is similar to the tutorial from http://wiki.ros.org/Robots/TIAGo/Tutorials/MoveIt/Pick_place, but with some important differences. I believe it would be nice in the introduction section to compare the differences between another monocular pick-and-place solution and the proposed one.
Why do the authors compare their results only with MLP? They should compare the proposed work with at least other monocular aproach in order to emphasize the advantages and restrictions of their work.
More general comments and minor errors are listed as follows.
"camera system." -> "camera systems."
"Active depth camera, e.g. time-of-flight (TOF) cameras that send continuous laser pulses to the target and then calculate the roundtrip time of the laser pulses to get the exact distance to the target, when TOF cameras send laser pulses towards the smooth surfaces with mirror reflection, only a few percentages of the pulses could return and received, making it difficult for the TOF camera to obtain accurate observations in this case." -> please rewrite
"cameras rely" -> "cameras, rely"
"pick-and-place, The" -> "pick-and-place. The"
"Convolutional Neural Networks (CNN)" -> the term was already defined in the text
"The on-policy algorithms modifies" -> "The on-policy algorithms modify"
"by parameterize" -> "by parameterizing"
"To limit the different" -> "To limit the difference"
"TRPO introduce" -> "TRPO introduces"
Figure 3 is not referenced in the text.
"We create 30" -> "We created 30"
"of the sub-environment" -> "of sub-environments"
Author Response
The response is in the file.

Round 2
Reviewer 2 Report
Congratulations! I'm satisfied with the current version of the paper and I believe the paper can be accepted now. Good job!
Please pay attention on some small fixes after submiting the final version:
"cameras utilize" -> "cameras, utilize"
"Lee et al. [18] present " -> "Lee et al. [18] presented "